# Spatial-Temporal Relationship between Water Resources and Economic Development in Rural China from a Poverty Perspective

**DOI:** 10.3390/ijerph18041540

**Published:** 2021-02-05

**Authors:** Zhaorunqing Liu, Wenxin Liu

**Affiliations:** College of Economics and Management, Northwest A&F University, Taicheng Road, Yangling 712100, China; liuzhaorunqing@nwafu.edu.cn

**Keywords:** water–economic poverty, harmonious development model, OECD model, integrated weight, spatial-temporal analysis, water–economic management

## Abstract

Guaranteeing sustainable development is a pressing issue in China. To this end, balancing economic development and the protection of limited water resources enables healthy and orderly economic development. This study details the application of a water poverty index and sustainable livelihoods approach using 25 indicators to evaluate the water situation and the economic situation in rural China from 1997 to 2019. The analysis results suggest the need for location-specific policy interventions. In addition, we determined whether the water poverty and economic poverty or their spatial types featured the phenomenon of agglomeration. This study also proposes a harmonious development (HD) model and found a significant relationship between water poverty and economic poverty. Next, we adopted a spatial and temporal perspective to analyze the causes of variation in HD level using the modified Organization for Economic Cooperation and Development (OECD) model and defined four HD levels using a classification method. The results revealed that the overall HD level was higher in the east than in the west. In conclusion, water poverty is associated with economic poverty; thus, there is a need for water and economic assistance strategies in pro-poor policies. The research findings also serve as a theoretical foundation for policies aimed at resolving conflicts between water use and economic development in rural China.

## 1. Introduction

Natural resources are the basis of a country or region’s economic growth. However, regions rich in natural resources do not necessarily have the fastest economic growth and may even be experiencing shrinking [1]. Thus, the relationship between the economy and natural resources has become a growing concern for governments and scholars [2]. There is a strong understanding that water is one of the most stressed resources, and it plays an increasingly important role in poverty alleviation and economic development in the world [3]. Water shortage is both a cause and a consequence of poverty [4]. In most areas of the world, water is central to poverty, and thus its provision is central to poverty alleviation, which is why this paper begins with a brief discussion of how water and poverty are interconnected [5]. Rapid population growth increases the domestic, agricultural, and industrial water demands [6], and improper planning and management further contribute to water shortages, which restricts economic development [7]. The lack of access to safe water hinders improvements in the quality of life and poverty alleviation [8]. Many people lack the ability to access water resources, and as a result they must spend more time, income, and other resources to meet their basic water demands. The resulting water resource shortage limits further development [9]. Conversely, hindered economic development also impedes the reversal of a water resources shortage situation [10]. Therefore, to better understand the relationship between water resources and economic development, understand alleviate water shortages, formulate economic poverty strategies, and determine the causes of these situations, the water shortages levels inevitably need to be measured.

Therefore, solving the issues of water resource shortage at multiple scales requires a multidimensional perspective. Sullivan proposed the water poverty index (WPI) [11]. Water poverty theory draws on poverty theory and organically combines the development, utilization, and management of water resources with people’s abilities to use water resources and environmental impact problems, thus creating an understanding and a unique approach to solving water shortages. This framework can serve as a theoretical basis for integrated water resources management, and through the integrated management, achieve the sustainable development of water resources and the economy [12]. Water problems are intertwined with socioeconomic and various other dimensions, which increasingly show diversity in complex combinations. The existing literature lacks research on the relationship between water and economic poverty. For example, Sullivan adopted a correlation coefficient method based on WPI and human development index (HDI), although the HDI can only reflect the per capita gross domestic product (GDP), life expectancy, and literacy, among other indicators, which do not fully elucidate the meaning of poverty [13]. Poor areas have been identified on the basis of per capita net income estimated using a rural index, which includes the per capita GDP and local income, suggesting the continued dependence on economic dimensions [14]. However, in addition to income and consumption, poverty includes a lack of access to opportunities, social services or exclusion, risk or vulnerability, and social deprivation [15]. Moreover, although the international community has made some progress in multidimensional poverty measurements, the selection and integration of a multidimensional index remains a key problem. The selection of metric dimensions and indicators are based on the characteristics of the poverty survey of basic needs, existing research experience in poverty and related correlation indexes, definitions of poverty, objectives of poverty reduction, and research frameworks [16]. An influential poverty measure is the sustainable livelihoods approach (SLA) proposed by the Department for International Development (DFID, U.K.) to establish an analysis framework for vulnerability and sustainable livelihoods [17]. Sharp’s [17,18] model is commonly used by researchers, given its simple and comprehensive characteristics.

The problem of water poverty and economic poverty concerns multiple systems, such as water resources, as well as the economy, society, and environment [12], whereas the WPI and the SLA use a single scale. The current study is limited to the simple interactive coupling of water poverty and economic poverty; thus, it is possible that relevant important information has been lost given the limited space and that the relationship between the issues of economic and rural water poverty is neglected. Therefore, it is necessary to combine the water resources and economic development for analysis. Further insight into the problem of water resource shortage can be gained by taking a comprehensive approach that analyzes the spatial and temporal variability in water resources and economic development. The objective of this study is to assess water shortages and economic development in rural China by applying two methodological frameworks that handle these limitations. We believe that the unique method presented in this article will serves as valuable reference for water resources studies in other countries. Our framework uses the WPI and SLA as the starting point and integrates both water poverty and economic poverty to analyze their relationship using the harmonious development (HD) model. The design of the WPI is based on the SLA framework, which allows for a theoretical comparative analysis [11]. We propose that the present method will serve as a valuable reference for studies on water resources and economic poverty in other countries, and the findings have theoretical and practical significance for the alleviation of water and economic poverty.

## 2. Study Area

In general, China’s gross domestic product (GDP) and water resource quantity decrease from southeast to northwest. As the most populous country in the world, most of the population is concentrated in the eastern, coastal, and central regions and dependent on industry and agriculture-based livelihoods (Figure 1). However, China is also a dry country that suffers from serious water shortages. In 2014, China’s per capita water resources were 2100 m^3^, which less than a quarter of the world’s average, and the temporal and spatial distribution remains uneven [18]. About half of North China’s population faces a severe water shortage, with per capita water resources of only 990 m^3^ [19]. Owing to the unreasonable development and utilization of water resources, as well as the failure to control water pollution and implement water protection measures, China is facing serious problems, including water scarcity, abuse, and pollution. To strengthen the management of the water resources and water facilities, water conservancy is indispensable to modern agriculture and the foundation of socioeconomic development. Water conservancy can help China improve its agricultural production and the income of poor farmers in particular. In addition, China is the world’s largest developing country, and at the same time, its government is faced with a severe poverty problem and the threat of a non-traditional poverty–water problem, which cannot be resolved using traditional economic measures [19]. Poverty has been prevalent throughout the history of social development, and particularly so in developing countries. In China, the problem of poverty is common in its rural areas, which the government has attempted to address with a series of medium- and long-term plans and development policies. However, according to large-scale household survey data estimating the incidence of the rural poor, China’s rural poverty problem is still very serious, and alleviating these levels has become increasingly difficult [20]. For example, in 2013, although the country reported the fastest annual GDP growth, the absolute number of rural poor in China increased compared to the previous year. Although rural China is rich in resources, its poverty levels are rather serious and have received much attention from the government.

## 3. Methodology

### 3.1. Model

#### 3.1.1. The Water Poverty Index

If WPI and SLA are used to evaluate China’s rural water resources and economic development, the actual situation of China must be considered more comprehensively. Based on the previous research results and the research perspective of this paper, the selection of indicators for rural water resources and economic development in this paper mainly follows the principles of scientificity, accessibility, and comparability, making the indicators more in line with the actual situation of rural water resources and economic development in China.

The methodology adopted in this study was based on the WPI [21], which evaluates the extent of water shortage using five components: resources (R), access (A), capacity (C), use (U), and environment (E). Resources indicate the physical availability and reliability of groundwater and surface water. Access is the prevalence of tap water and irrigation; this component accounts for the demand of water for basic functions, as well as for agriculture and sanitation, and reflects the extent of the public’s proximity to clean and safe water. Capacity indicates the water management ability and is based on aspects such as the education, health, and financial situation of the population. The component reflects the influence of one’s socioeconomic status on water resources. Use denotes the water use efficiency in the domestic, industrial, and agriculture sectors. Finally, environment is the environmental status as related to water resources management, including the potential pressure of the ecological environment on water quality (Table 1). The five WPI components are set at the same weight, as shown in the following equation:(1)WPI=0.2×Resources+0.2×Access+0.2×Capacity+0.2×Use+0.2×Environment

#### 3.1.2. Sustainable Livelihoods Approach

Scoones applied the sustainable livelihoods approach (SLA) to livelihood activities across poverty reduction on the people oriented [26]. He combined five components—natural capital, physical capital, financial capital, human capital, and social capital—with factors of production to identify opportunities for the continuous growth in social development. The factors complement each other between the different types of capital and do not replace them. A community is defined as impoverished when it lacks the five types of capital. Table 2 summarizes the SLA components, indicators, variables, data sources, and references used in this study.

The five types of capital and their individual components are as follows:(i)Natural capital: sunshine, clean air, land, water, forest, and minerals;(ii)Physical capital: machines, factories, tools, equipment, and facilities;(iii)Financial capital: credit, savings, and remittances;(iv)Human capital: education, knowledge, skills, training, health spending, and migration;(v)Social capital: social relationships in the market, wealth, power, prestige, and social networks.

The five SLA components are set at the same weight according to the following equation:(2)WPI=0.2×Financial+0.2×Human+0.2×Natural+0.2×Physical+0.2×Social

#### 3.1.3. The Harmonious Development Model

Economic poverty and water poverty complement each other. To pursue economic development, economic poverty is often improved at the expense of water poverty. Ideally, economic poverty and water poverty should be improved simultaneously. To calculate the harmonious and developmental abilities between economic poverty and water poverty, we drew a rectangular coordinate diagram that shows the possible harmonious degree and developmental conditions of the economic poverty and water poverty in China. In the figure, the *y*-axis represents the value of rural economic poverty, and the *x*-axis represents the value of rural water poverty. The dotted curve y = x^1/3^, y = x, and y = x^3^ divides the square area from the upper left to the lower right into four equal parts as a metric of a harmonious ability (Figure 2). The improvement in economic and water poverty should show a harmonious tendency. The closer to y = x, the higher the harmonious ability [30]. The developmental function is used to calculate improvement ability of economic poverty and water poverty. The solid curves y = ½ − x^3^, y = ¾ − x^3^, and y = 1 − x^3^ divide the square area from the lower left to the upper right into four equal parts as the metrics of developmental ability. As the solid line extends outward, development ability increases. This function is based on the production possibility curve and reflects the non-linear interactive relationship between economic and water poverty conditions. The developmental function reflects the developmental ability of the economic and water poverty. The closer the value is to 1, the better the developmental ability. Hence, a harmonious and developmental tendency in the improvement of economic and water poverty can be shown by combining the two models.

Based on the connotation of harmonious degree (H) and development degree (D), the model is calculated using the following equations:y = x^a^, H = a(a < 1), H = 1/a(a > 1),(3)
y = D − x^3^(4)

Hence, the harmonious development model (HD) of economic and water poverty systems includes both H and D. It is calculated using the following equation:HD = D^a^ ∗ H^b^.(5)

China, as the largest developing country in the world, has maintained a rapid development pace. China’s Twelfth Five Year Plan, issued by the State Council (2012), set development as an important goal. However, along with economic development, increasing attention is being devoted to the development of social harmony. Therefore, equal weight is assigned to H and D. Thus, the final HD is the approximate weight value of H and D, with both weights set to 0.5.

#### 3.1.4. Modified OECD Model

The Organization for Economic Cooperation and Development (OECD) model is based on changes in variation [31]. This model changes the relative amounts and the considerations made by the two comprehensive systems. The period of the variational analysis reflects the relationship between the systems, which further improves the objectivity and accuracy of the driving factors:(6)$$D_{f}=\Delta WP/\Delta EP$$
where ΔWP is the variation in water poverty, and ΔEP is the variation in economic poverty. This study adopted the rate of change for leading driving factors between the two systems.

The OECD model has the following advantages. Firstly, it provides stable results; and secondly, it is not affected by statistical dimensional changes, a feature lacking in statistical methods and systems specific to China [32]. Thirdly, it can be used to clarify the factors contributing to systemic changes and develop reasonable reduction policies and improve the OECD model in the context of China. The modified OECD model can more accurately reflect the water and economic poverty across different areas and time intervals in rural China. The trend of economic growth in China has maintained its high speed, while the rural economy continues to exhibit a continuous economic growth rate. Integrating the above discussion, this study only considered ΔEP>0. When ΔWP/ΔEP<−1, the region has achieved rapid economic development as well as reduced water poverty, suggesting a progressive interaction between the two systems or a strong water poverty lag. When 0.5<ΔWP/ΔEP<1, there was an annual improvement in the water poverty along with economic development, although the economy grows faster than the improvements in water poverty, indicating a weak water poverty lag. When ΔWP/ΔEP>1, economic growth lagged far behind improvements in water poverty; this is known as a strong economic poverty lag (Figure 3).

### 3.2. Assigning Weights to the Indicators

The assignment of integrated weight affects the reliability and accuracy of the final results, which in turn influences decision-makers referencing the results in their management decisions. Past studies had two approaches to assigning weights: assigning equal relative weights and different relative weight [33]. To calculate the most reliable results, this study combines the two methods. In the process of ascertaining the weight of the variables, the importance of both weights was different, while in the five components, the importance of both weights was the same. Furthermore, we applied a multi-criteria decision-making (MCDM) method to determine the indicator weight. In general, many researchers have applied the subjective and objective weighting methods to improve decision-making [34]. Subjective methods, such as Delphi and the analytic hierarchy process (AHP), determine weights on the basis of expert experienced judgment and can reflect the specific situation of indicators; however, they do not reflect their economic and technical significance. Objective weighting methods, such as entropy, criteria importance, inter-criteria correlation and (the Technique for Order Preference by Similarity to an Ideal Solution) TOPSIS, are based on the analysis of measurable data. The results may not be from analysts; therefore, the importance assigned to the weights may differ. Based on the consultation of experts and their experience, AHP is an effective and widely applied method in assigning weights and plays a crucial role in the evaluation and analysis of indicators. The entropy method of weighting allows for the consideration of an ideal water poverty situation [35]. Therefore, to systematically assign weights to indicators, this study combined two MCDM methods, AHP and entropy. Integrated weights combined AHP and entropy to highlight the importance of each indicator [36] using the weighted synthesis of the WPI and SLA values. The optimization model determining the integrated weight of the indicator was established using the least squares method. This method is suitable for complementary and uncertain information and can transform this subjective uncertainty and complex information into deterministic decision results. This study draws on the results of existing research that combined subjective and objective weighting methods. When using the method, the subjective weight synthesized uncertainty, determined the objective weights, and improved the accuracy of the weights.

The AHP-determined subjective weighting vector *v* is defined as:(7)$$v={\left(v_{1},v_{2},\ldots ,v_{m}\right)}^{T}$$

The entropy-determined objective weighting vector *u* is defined as:(8)$$\begin{array}{c} u={\left(u_{1},u_{2},\ldots ,u_{m}\right)}^{T}\\ v={\left(u_{1},u_{2},\ldots ,u_{m}\right)}^{T} \end{array}$$

Finally, the integrated weighting vector is:(9)$$\begin{array}{c} w={\left(w_{1},w_{2},\ldots ,w_{m}\right)}^{T}\\ v={\left(w_{1},w_{2},\ldots ,w_{m}\right)}^{T} \end{array}$$
where v is the AHP weighting, u is the entropy weighting, T is the vector, and w is the integrated weighting.

For all sample cases, the error of the integrated weighting evaluation should be as small as possible. The least squares minimization problem, using the integrated weight $w_{j}$, is given by:(10)$$minH\left(w\right)=\sum_{i=1}^{n}\sum_{j=1}^{m}\left\{{\left[\left(u_{j}-w_{j}\right)z_{ij}\right]}^{2}+{\left[\left(v_{j}-w_{j}\right)z_{ij}\right]}^{2}\right\}$$
(11)$$\sum_{j=1}^{m}w_{j}=1,\;w_{j}\geq 0\;\left(j=1,2,\ldots ,m\right)$$
where n is the number of indices i, m is the number of indices j, and $z_{ij}$ is the normalized matrix. The optimization model was solved by constructing a Lagrangian function [30]. The results are displayed in Table 3.

## 4. Results and Discussion

### 4.1. WPI and SLA Results for Rural China and Their Significance

The water and economic poverty values in rural China were calculated as follows. Firstly, the indicators corresponding to different measurements of raw data were addressed by data standardization. Secondly, the weight of each indicator was determined by the integrated weight method, and then the water and economic poverty values in rural China were calculated by the weighted summation to obtain the comprehensive evaluation value for each component using its indicators. Finally, we combined the average value with a specific analysis using clustering (which is mainly used for information management and decision analysis, classified according to the individual characteristics of the research object) as a standard to elucidate the actual situation and analyze the causes.

As noted in the methods section, we obtained the total water and economic poverty values in rural China on the basis of each component of WPI and SLA by applying the weight summation method. The final evaluation results reflect the actual water and economic poverty situations in rural China for 1997–2019. According to the results, the water poverty value ranged from 0.229 to 0.553, and the economic poverty values ranged from 0.093 to 0.572 in rural China during 1997–2019. The obvious change in values in Table 4 reveals the degree of improvement in the water and economic poverty situations in rural China.

Due to space limitations, we have only listed certain years. W is the water poverty level; E is the economic poverty level. Water poverty is not improving as fast as the economy poverty, although it can be said that the water and economic poverty of 31 provinces in rural China is gradually improving. However, the absolute difference between water and economic poverty values between coastal and inland areas in rural China has gradually widened, indicating that the improvements in the coastal and inland water poverty situation are not harmonious. For example, Guangdong lies in the southeast coastal area. Its water poverty values increased from 0.415 to 0.518, with an average annual growth rate of 1.13%. However, the economic poverty values in Guangdong increased from 0.246 to 0.638, with an average annual growth rate of 7.24%. The economic development speed is much faster than the improvement speed of water resources system. In the inland region, the water poverty in Shaanxi increased from 0.229 to 0.272, with an average annual growth rate is 0.85%, which was almost at a standstill. Policy intervention on water resources system was urgently needed. However, the economic poverty in Shaanxi increased from 0.083 to 0.331, with an average annual growth rate of 13.58%. The economic development speed is much faster than the improvement speed of water resources system, which showed a situation of disharmony between the water resources and economic development. We also found that the water and economic poverty of 31 provinces showed gradual agglomeration, although we leave a further analysis of this relationship to future research.

These findings have extensive implications for improvements in water resource shortages and economic poverty, which may lead to inefficient investment and limit their own conditions at any scale [37]. In addition, it appears important to give preferential policy treatment to certain areas that have limited resource availability, low socioeconomic levels, and both water and economic poverty. Admittedly, models developed using data representing average conditions do not represent the actual conditions in a given location. However, some rules based on average values can serve as a better guide for improvements in the actual situation. We incorporate the average value in a specific analysis using clustering to distinguish and observe the actual situation and examine the underlying causes. According to the regional value of water and economic poverty, we then conducted a clustering analysis using SPSS [19] to assess each province in rural China.

The cluster analysis revealed four degrees of water and economic poverty in rural China: low, medium, severe, and very severe. The spatial distribution of the water poverty and economic poverty in the coastal regions is superior to that in the inland regions (Figure 4). Water poverty and economic poverty refer to water shortages level and water development level. Low water poverty and economic poverty indicates a good situation, and severe water poverty and economic poverty indicates a bad situation.

### 4.2. WPI and SLA Component Results for Rural China and Their Significance

In general, the WPI and SLA values provide implications for water resource management and poverty alleviation in Table 5 [38] In addition, in giving preferential policy to areas with socio-economic levels, resource availability conditions may result in failure toward alleviating water poverty and economic poverty [39]. With the WPI component, resources (R), access (A), capacity (C), use (U), environment (E) and SLA components financial capital (F), human capital (H), natural capital (N), physical capital (P), and social capital (S), this study clearly shows that specific policies should be formulated. The component values help prioritize focus areas in the relevant study area as well as to monitor the degree of water shortages and poor to improve in the specific focus areas. For example, in Ningxia, resources and access components of water poverty, human capital, natural capital, as well as the physical capital components of economic poverty, should be addressed as a priority due to their values being the lowest compared to other WPI and SLA components. Additionally, within resources and use components, policy should focus on improving the situation of numbers of reservoirs and the percentage of the rural population with access to clean water, because some of the components, indicators and variables cannot be managed (e.g., the resource variability and availability) [40]. Beneficial development policy has been adopted (e.g., increasing water supply and increasing investments in sanitation) when considering the actual economic situation in the region. Improvements in the water resources shortage situation might be easier if the focus was on increasing economic development ability; thus, controlled effective policy would be necessary [41].

### 4.3. Temporal and Spatial Variation between Water Poverty and Economic Poverty

Table 6 and Figure 3 reflects the HD and lag levels of water and economic poverty using the HD and modified OECD model. Using decoupling data, combined with HD model results, we can further determine which factor is backward between urban areas and rural areas. A higher HD indicates that the coordination level is higher, suggesting a harmonious and developmental relationship between the better use of water resources and poverty alleviation. On the other hand, hindrances in development can lead to a vicious circle. The lag in decoupling water and economic poverty gradually reduced from the east to west, reflecting the real situation in China. To eliminate the impact of cyclical fluctuations in data, we examined the significance of the water poverty values and economic poverty values. Standards applicable to this study focused on the means of the HD model (Table 6) and were used to identify the reasons underlying the discrepancies from a temporal and spatial perspective at a provincial level. For example, the HD level of Beijing was 0.479, which had a strong water lag in 1997–2003; in 2012–2019, the HD level of Beijing was 0.496, also with a strong water lag. This shows that the contradiction between economic development and water shortage has been alleviated; however, water shortages remain a hindrance. By contrast, the HD level of Xizang was 0.185, and it had a strong water lag in 1997–2003; in 2012–2019, the HD level of Xizang was 0.226, and it had a strong economic lag. This shows that the contradiction between economic development and water shortage has been serious. Reasons for the inharmonious relationship are needed; this suggests policy intervention. 

To facilitate related decision-making, the HD level of water and economic poverty in rural China can be divided into four categories on the basis of the clustering analysis method of SPSS [19]:

(1) Strong HD ability: This category includes the regions of Shandong, Jiangsu, Shanghai, Zhejiang, and Guangdong. Most of these areas are at the forefront of China’s reform and opening up economic development, and standard of living and rank the highest in science, education, culture, and healthcare. In addition, these regions have a well-developed water system, abundant rainfall, economic development, and less pressure on water resources; however, they need to improve awareness regarding ways to save water and protect the environment. Finally, both water and economic poverty are at the national minimum and both promote and complement each other.

(2) Medium HD ability: This category includes the regions of Heilongjiang, Jilin, Liaoning, Neimenggu, Beijing, Tianjin, Shanxi, Shanxii, Hebei, Henan, Hubei, Hunan, Sichuan, Chongqing, Anhui, and Fujian. Of them, Heilongjiang, Neimenggu, Liaoning, and Jilin were in the weak HD category during 1997–2011. Heilongjiang showed a strong water and economic poverty lag. Water and economic poverty in Heilongjiang, Jilin, Liaoning, and Neimenggu were at the medium level. The level of natural water resources and economic development were lower than the eastern coastal area. In addition, there was low pressure on water but a severely damaged ecological environment, low water-saving consciousness, large agricultural production, and low use efficiency of water for agricultural purposes; thus, water poverty levels showed moderate deviation. The poverty evaluation revealed the need for improvements in income, education, and medical levels, with most indicators needing neutralization. Thus, water and economic poverty fell within the medium range.

During the period 2004–2011, Beijing, Tianjin, and Hebei showed a weak HD level; although these regions have poor water resources conditions, they demonstrated strong social adaptation ability, through improved infrastructure and the utilization efficiency of water resources. This contributed toward reduced water shortages, normal production in the national economy and living needs, and relatively low water poverty. From 1997 to 2011, Henan reported medium HD levels, which improved in 2012–2019 along with the water poverty lag. As China’s most populous province, a large number of men have relocated seeking employment opportunities left elderly women and children behind, which has led to a lower rural water capacity. Nevertheless, despite its low economic levels, Henan has good regional conditions in terms of water resources, thus contributing to the momentum in the eastern coastal developed regions and inland radiation diffusion zone, social adaptation, water drainage facilities, government regulation, control ability, and improved water use efficiency. The evaluation of poverty evaluation, provincial indexes, average rural per capita income, and education level reveal a higher poverty level because of the economic structure, accumulation ability, scientific and technological strength, and imperfect market mechanism. This suggests that reducing the incidence of poverty can help improve people’s ability to adapt to water shortages and alleviate water poverty, and doing so will upgrade these regions to the medium HD level.

Anhui, Fujian, Hunan, and Hubei reported better water resource conditions, but low economic levels. The evaluation further revealed that poverty in these regions can be attributed to economic structure, accumulation ability, scientific and technological strength, and imperfect market mechanism. Shanxi and Shaanxi showed dual effects on water resources and ecology population, as well as low annual precipitation, water use efficiency for agricultural purposes, and economic development. The ineffective protection of environment further aggravated by severe crowding in terms of water usage has led to water loss and pollution and soil erosion. Chongqing and Sichuan have good water conditions, low pressure on water, and no damage to the ecological environment, but low economic development, self-sufficiency in agricultural facilities, government and social urban poor performance, and water-saving awareness. Thus, with poor social adaptation and water poverty, these regions fall within the medium HD category. In terms of poverty, revenue expenditure in education, healthcare, and environmental development in these areas are not optimistic; moreover, they are characterized by underdeveloped infrastructure and low water resource use efficiency.

(3) Weak HD ability: This category includes the regions of Guangxi, Guizhou, Yunnan, Xinjiang, Jiangxi, and Hainan. Of these, Guangxi, Guizhou, and Yunnan are located in the southwest of China, which has a subtropical humid climate. The pressure on water was low and there was no damage to the ecological environment; however, the regions had low economic development, self-sufficiency in agricultural facilities, government and social urban performance, and water-saving awareness. Thus, regions with poor social adaptation to water poverty fall under the medium HD level. The poverty evaluation revealed that revenue expenditure for education, healthcare, and development were not optimistic in these areas; in fact, poverty has contributed to underdeveloped infrastructure and low use efficiency of water resources. Figure 5 reveals no obvious lag in 2012–2019. In particular, Xinjiang had a weak water poverty lag. In 1997–2003, Xinjiang was in a weak HD state. Furthermore, despite minor improvements, Xinjiang had a poor water resource situation and restricted economic development and poverty alleviation. In addition, the poor condition of natural water resource conditions and low social adaptation ability has affected water alleviation measures.

The poverty evaluation for Xinjiang revealed low-income levels in urban and rural regions, education investment, employment, and coverage of highway construction. Nevertheless, the economic poverty level was better than that of water, although the lack of water resources has restricted poverty alleviation measures and economic development. Hainan, a rural economy, showed improvements in its water poverty conditions in 2004–2011, with the water and economic poverty rates having no effect on development in 2012–2019. Hainan is located in the south of China, and although it has high temperatures and rains throughout the year, it is surrounded by the sea and lacks freshwater resources. China’s winter fruits and vegetables require large amounts of freshwater resources. Despite its self-sufficiency and local finance, the implementation of water-saving irrigation and water conservancy facilities has faced several delays. In addition, the rate of pollution treatment decreased compared to that of chemical fertilizer applications, with economic losses arising from the per capita water pollution. The excess demands, insufficient supply of water resources, and low economic development have resulted in weak HD levels for water and economic poverty.

(4) Very weak HD ability: This category includes Gansu, Ningxia, Qinghai, and Xizang. Gansu and Ningxia have a high degree of natural water poverty and low socioeconomic development, use efficiency of water for agricultural purposes, and self-sufficiency in terms of finance, production capacity, and science and technology. In addition, the education levels in these regions have further contributed to the seriousness of water poverty. The poverty evaluation revealed low-income levels; maternal and child healthcare; and medical, transportation, and communication facilities, and a high Engel coefficient and illiteracy rate. Thus, water and economic poverty are two serious issues closely related to each other. Xizang and Qinghai have a good water resource background; however, their economic development is low, and the development dynamics, social production, and government regulation are insufficient. Thus, the ability to socially adapt, the performance of water and drainage facilities, use efficiency of water resources, and water-saving consciousness are also low. The poverty evaluation suggests that urban and rural income in these areas, as well as science, education, culture, and health, demonstrate moderate deviations. In addition, the coverage of social security, income of urban and rural residents, education investment, health undertakings, and employment rate are low, and traffic conditions, infrastructure, and water resource use efficiency are poor.

## 5. Summary and Conclusions

Establishing an index system for water and economic poverty specific to China will not only effectively reflect the improved situation of water and economic poverty, but also compensate, to a certain extent, for the deficiencies in the existing index system for both water poverty types. This study proposed a framework that integrates the weighting method and combines two models to analyze the spatial and temporal relationship between water and economic poverty. The model provides comprehensive insights into both poverty types using WPI and SLA, which are considered holistic tools to assess a water resource shortage and economic poverty. In addition, the results obtained using the HD and OECD models offer implications for more effective water policies and to help policymakers better understand the linkages between water and poverty. This analysis set 25 indicators in the WPI and SLA to assess water and economic poverty in China, considering local issues and limited data availability. The results show that the water and economic conditions in China worsened from the east to west. The scores of each indicator suggest that water poverty is linked to poor water use and environmental integrity, which clearly indicates inappropriate water management rather than resource insufficiency. We then built quantitative models to describe the HD level in rural China and further identified the factors affecting water and economic poverty and the factors causing spatial variations. The indicator system for water poverty and economic evaluation currently used in China has too many indicators and does not prioritize the allocation of water and economic capacity. This was the main factor motivating the present study in proposing the HD model as an alternative. Although there are empirical studies on the relationship between water resources and economic development in academic circles, there are a lack of empirical studies on the coordinated development of water resources and economic development from the perspective of poverty. For a long time, the academic circles and the government have primarily judged or conducted case studies on issues, such as water resources constraining economic development, economic development restricting water resources improvement, the main characteristics of water resources and economic development, and the basis for policy formulation, but have not discussed the above issues from the perspective of empirical research. The research of this paper makes up the blind spot of empirical analysis in this field and verifies the theory and policy proposition that water resources constrain economic development and economic development constrains water resource improvement. The HD level of water and economic poverty in rural China showed harmonious and developmental improvement, emphasizing the continuous improvements of rural water and economic poverty as key tasks for the future. In addition, the water shortages and economic development based on WPI and SLA remains a preliminary study. There are a few problems that have yet to be researched. On the one hand, the data do not represent deviations from these averages at finer geographical or temporal scales, and the rationality of the data will be the focus of our next study. On the other hand, the reasonable selection of indicators will continue to be the focus of research. The conclusion of this paper is based on the literature and national conditions of the index selection. However, with the vast territory of China, there are obvious differences in the production factor capabilities and development levels between regions. With the development of the economy, the flow of population and the diffusion of technology will inevitably change the development level between regions in China dynamically. Therefore, the selection of different indicators may lead to great differences in the results.

These research findings serve as a theoretical foundation for policies aimed at relieving conflicts between water resources and economic development in rural China. It accounts for spatial changes affecting the relationship between both poverty types in rural China. The policy recommendations on the basis of our results are as follows. In considering the water rights and capacity, economic development, and other factors in rural China, the decoupling ability of water and economic poverty can be improved through targeted and operable policies. When the HD ability of a region is strong, these regions must adjust the degree of fair income distribution; at the same time, the fairness of water resource allocation should be strengthened, especially in East China where the balance is poor. Water resources within and between regions should be mobilized according to the principle of proximity. Based on regional ecological land area and water resource endowment, policies should be designed to improve water resource utilization efficiency, especially in the southern region; strengthen environmental protection; and maintain areas naturally rich in water resources by improving water-saving awareness. When the HD ability of a region is in the mid-range, these regions must continue to maintain the steady growth of the economy. Taking advantage of the national strategic opportunity of the coordinated development of the Yangtze River Economic Belt, China should improve the coverage of the regional transportation network, focus on supporting the development of high-speed rail and civil aviation, and enhance the interaction efficiency with economically developed regions; strengthen economic ties with developed eastern provinces; further improve their ability of social production and social security; prioritize the protection of water resources and the ecological environment; and promote people’s livelihoods. Finally, when the HD ability of a region is weak, the HD degree of water and economic poverty of these regions are not high. Assuming that the conditions of the local water resources cannot change, emphases should be on raising the level of socioeconomic development; all regions should strengthen water-saving aspects of propaganda and education to the public. In particular, areas with less water should encourage residents and enterprises to save water, improve water use efficiency, and the regions with rich water resources should strengthen the control of total water, put an end to wasting water, and radically reduce their water consumption. While optimizing the industrial structure, the provinces with low economic development level should reduce the proportion of agricultural industries with low water use efficiency and increase the proportion of secondary and tertiary industries with high water use efficiency to reduce the water resource consumption, ensuring that economic development is not achieved at the expense of the ecological environment. These areas can be developed by alleviating water and economic poverty by implementing effective policies and through sufficient financial support.

## Figures and Tables

**Figure 1 ijerph-18-01540-f001:**
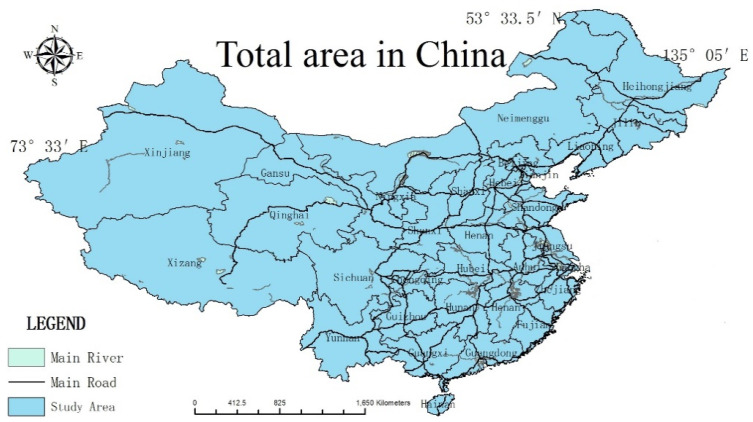
Study area in China.

**Figure 2 ijerph-18-01540-f002:**
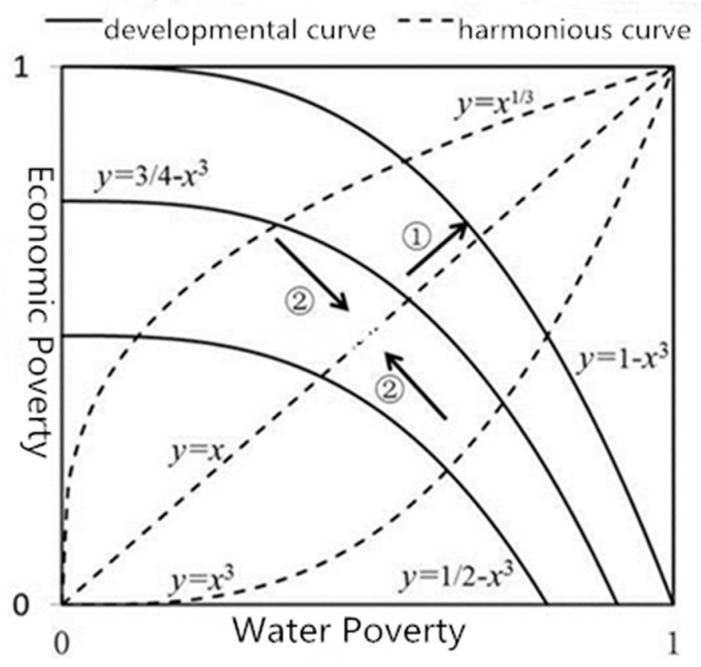
Rural water poverty and economic poverty harmonious development (HD) model.

**Figure 3 ijerph-18-01540-f003:**
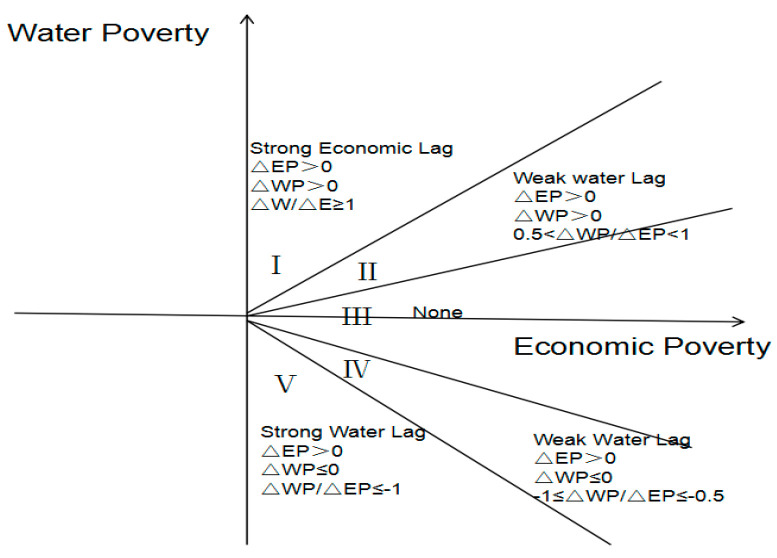
Driving analysis of water poverty and economic poverty.

**Figure 4 ijerph-18-01540-f004:**
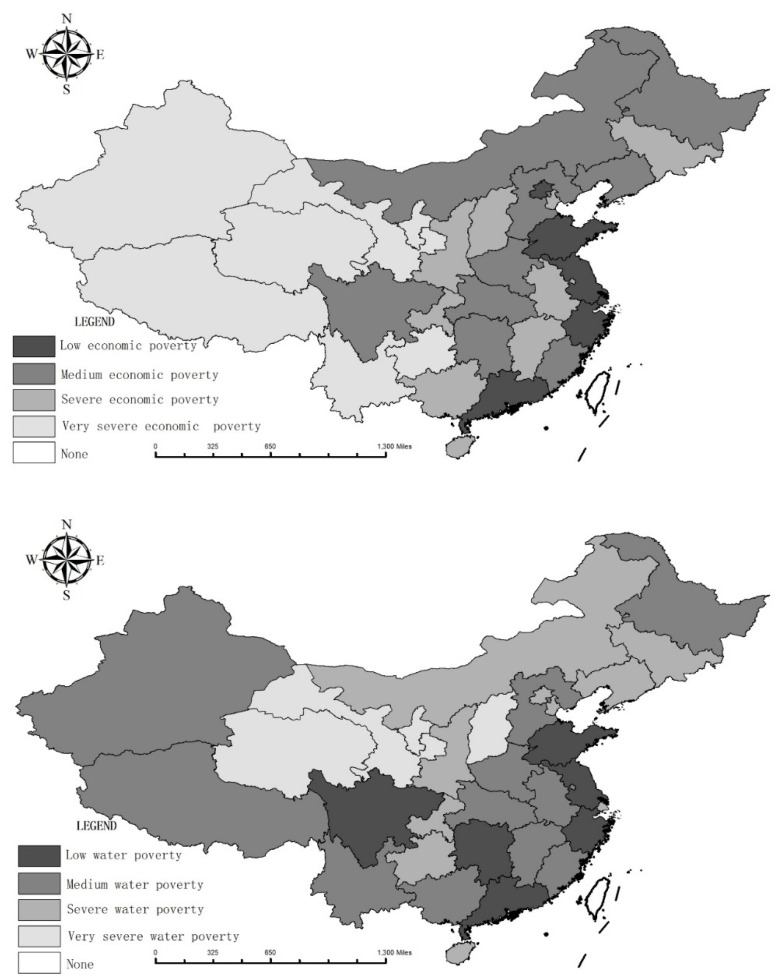
The spatial pattern of water poverty and economic poverty in China from 1997–2019.

**Figure 5 ijerph-18-01540-f005:**
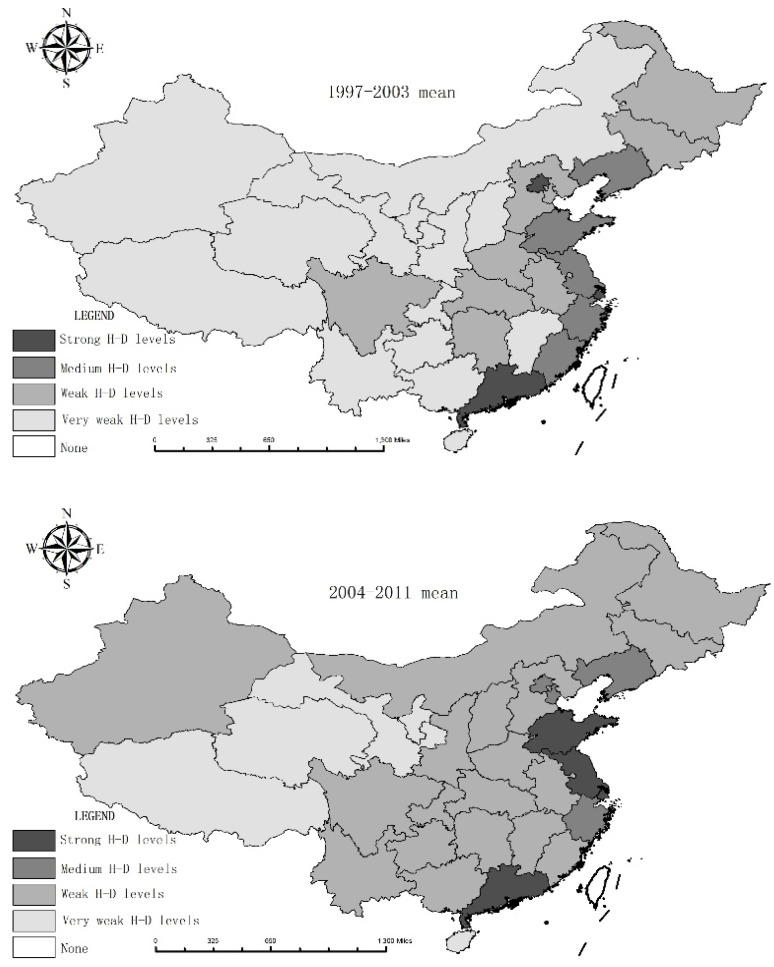
The temporal and spatial variation condition of H-D ability of water poverty and economic poverty in rural China.

**Table 1 ijerph-18-01540-t001:** Details of the water poverty index (WPI) components, indicators, and references.

Component.	Indicator	Relationship with Water Poverty	References
Resources (0.2)	Rainfall (R1)	High R1—Less water poverty	[21]
	Per capita annual rural water resources (R2)	High R2—Less water poverty	[21]
Access (0.2)	Numbers of reservoirs (A1)	High A1—Less water poverty	[22]
	Percentage of population with access to clean water (A2)	High A2—Less water poverty	[23]
	Actual irrigation capacity (A3)	High A3—Less water poverty	[21]
Capacity (0.2)	Per capita annual rural gross domestic product (C1)	High C1—Less water poverty	[22]
	Number of doctors per ten thousand people (C2)	High C2—Less water poverty	[24]
	Male migrant workers (C3)	High C3—High water poverty	[12]
Use (0.2)	Per capita per day rural domestic water use (U1)	High U1—Less water poverty	[25]
	Portion of water use for irrigated land (U2)	High U2—Less water poverty	[21]
Environment (0.2)	Chemical fertilizer use per hectare of cultivated area(E1)	High E1—High water poverty	[21]
	Soil and water loss control area(E2)	High E2—Less water poverty	[21]

Note: For example, R1 represents the first indicator of Resources.

**Table 2 ijerph-18-01540-t002:** Details of the sustainable livelihoods approach (SLA) components, indicators, and references.

Component.	Indicator	Relationship with Economic Poverty	References
Financial capital (0.2)	Per capita GDP (F1)	High F1—Less economic poverty	[27]
	Engel’s coefficient (F2)	High F2—High economic poverty	[28]
Human capital (0.2)	Illiteracy rate (H1)	High H1—High economic poverty	[26]
	Agricultural population (H2)	High H2—Less economic poverty	[29]
	Physicians per capita (H3)	High H3—Less economic poverty	[29]
Natural capital (0.2)	Average crop production (N1)	High N1—Less economic poverty	[26]
	Cultivated land per capita (N2)	High N2—Less economic poverty	[26]
	Rainfall (N3)	High N3—Less economic poverty	[27]
Physical capital (0.2)	Road mileage per capita (P1)	High P1—Less economic poverty	[26]
	Agricultural machinery per capita (P2)	High P2—Less economic poverty	[26]
	Electricity consumption per capita (P3)	High P3—Less economic poverty	[26]
Social capital (0.2)	Urbanization (S1)	High S1—Less economic poverty	[26]
	Level of social justice (S2)	High S2—Less economic poverty	[27]

Note: For example, F1 represents the first indicator of Financial capital.

**Table 3 ijerph-18-01540-t003:** Subjective weights, objective weights, and integrated weights of the WPI and the SLA components, indicators, and variables.

System	Component	Variable	AHP	Entropy	Integrated
WATER POVERTY	Resources (0.2)	Rainfall	0.333	0.401	0.446
		Per capita annual water resources	0.667	0.599	0.554
	Access (0.2)	Number of reservoirs	0.250	0.620	0.522
		Percentage of rural population with access to clean water	0.500	0.129	0.227
		The actual irrigation situation	0.250	0.251	0.251
	Capacity (0.2)	Per capita annual rural gross domestic product	0.493	0.501	0.400
		Elementary education enrolment rate	0.196	0.231	0.310
		Number of doctors per capita	0.311	0.268	0.290
	Use (0.2)	Per capita per day rural domestic water use	0.500	0.686	0.534
		Portion of water use to irrigated land	0.500	0.314	0.466
	Environment (0.2)	Chemical fertilizer use per hectare of cultivated area	0.667	0.190	0.429
		Soil and water loss control area	0.333	0.810	0.572
ECONOMIC POVERTY	Financial capital (0.2)	Per capita GDP	0.667	0.617	0.632
		Engel’s coefficient	0.333	0.383	0.368
	Human capital (0.2)	Illiteracy rate	0.311	0.472	0.400
		Agricultural population	0.493	0.347	0.410
		Physicians per capita	0.196	0.181	0.190
	Natural capital (0.2)	Average crop production	0.200	0.494	0.333
		Cultivated land per capita	0.400	0.289	0.352
		Water resources	0.200	0.217	0.315
	Physical capital (0.2)	Road mileage per capita	0.333	0.331	0.332
		Agricultural machinery per capita	0.333	0.226	0.280
		Electricity consumption per capita	0.334	0.442	0.388
	Social capital (0.2)	Urbanization (S1) +	0.500	0.612	0.589
		Level of social justice (S2) +	0.500	0.388	0.411

**Table 4 ijerph-18-01540-t004:** Calculated WPI and SLA values in rural China from 1997 to 2019.

W/E	1997	2003	2008	2013	2019	Mean
Beijing	0.206/0.216	0.208/0.283	0.238/0.422	0.285/0.553	0.279/0.632	0.234/0.382
Tianjin	0.171/0.166	0.195/0.197	0.213/0.287	0.278/0.404	0.245/0.464	0.212/0.272
Hebei	0.289/0.159	0.307/0.172	0.323/0.244	0.345/0.340	0.356/0.375	0.321/0.235
Shanxi	0.204/0.105	0.207/0.113	0.219/0.180	0.244/0.252	0.255/0.283	0.220/0.168
Neimenggu	0.239/0.101	0.251/0.117	0.274/0.205	0.315/0.324	0.356/0.368	0.275/0.193
Liaoning	0.216/0.170	0.237/0.182	0.256/0.257	0.288/0.381	0.289/0.430	0.252/0.255
Jilin	0.238/0.125	0.236/0.135	0.272/0.182	0.319/0.254	0.318/0.294	0.267/0.178
Heilongjiang	0.312/0.152	0.333/0.152	0.336/0.194	0.396/0.265	0.423/0.301	0.350/0.195
Shanghai	0.228/0.241	0.255/0.290	0.262/0.414	0.318/0.485	0.339/0.515	0.267/0.368
Jiangsu	0.363/0.223	0.393/0.259	0.427/0.399	0.410/0.565	0.423/0.642	0.400/0.373
Zhejiang	0.327/0.173	0.364/0.219	0.396/0.352	0.394/0.469	0.421/0.511	0.378/0.313
Anhui	0.333/0.128	0.348/0.132	0.362/0.181	0.396/0.265	0.422/0.303	0.366/0.191
Fujian	0.336/0.161	0.339/0.170	0.374/0.238	0.398/0.330	0.429/0.375	0.366/0.228
Jiangxi	0.366/0.116	0.356/0.116	0.351/0.166	0.378/0.241	0.403/0.276	0.363/0.162
Shandong	0.357/0.213	0.372/0.245	0.397/0.367	0.427/0.494	0.429/0.561	0.389/0.340
Henan	0.309/0.163	0.345/0.174	0.369/0.247	0.407/0.339	0.396/0.380	0.362/0.236
Hubei	0.310/0.142	0.335/0.152	0.346/0.211	0.375/0.308	0.390/0.370	0.344/0.211
Hunan	0.404/0.134	0.453/0.143	0.413/0.205	0.421/0.294	0.453/0.336	0.420/0.199
Guangdong	0.415/0.246	0.413/0.299	0.415/0.444	0.492/0.576	0.518/0.638	0.436/0.403
Guangxi	0.339/0.109	0.332/0.114	0.329/0.162	0.358/0.230	0.380/0.264	0.340/0.158
Hainan	0.210/0.082	0.227/0.097	0.252/0.133	0.296/0.188	0.298/0.215	0.247/0.131
Chongqing	0.253/0.105	0.259/0.117	0.267/0.174	0.281/0.259	0.284/0.298	0.263/0.168
Sichuan	0.354/0.142	0.359/0.150	0.374/0.218	0.398/0.313	0.429/0.362	0.381/0.212
Guizhou	0.257/0.065	0.249/0.084	0.259/0.119	0.271/0.167	0.274/0.211	0.257/0.116
Yunnan	0.341/0.101	0.330/0.104	0.339/0.147	0.340/0.204	0.357/0.244	0.336/0.144
Xizang	0.347/0.059	0.361/0.044	0.323/0.080	0.331/0.124	0.332/0.148	0.338/0.080
Shaanxi	0.229/0.083	0.231/0.113	0.237/0.168	0.269/0.272	0.272/0.331	0.243/0.169
Gansu	0.189/0.070	0.200/0.088	0.217/0.123	0.243/0.170	0.244/0.207	0.213/0.119
Qinghai	0.200/0.044	0.214/0.073	0.219/0.111	0.228/0.166	0.220/0.202	0.216/0.108
Ningxia	0.152/0.059	0.175/0.071	0.187/0.119	0.203/0.182	0.215/0.215	0.185/0.114
Xinjiang	0.292/0.100	0.334/0.117	0.329/0.143	0.325/0.205	0.341/0.242	0.328/0.145

**Table 5 ijerph-18-01540-t005:** WPI and SLA components values in rural China.

W/E	R/F		A/H		C/N		U/P		E/S	
	W	E	W	E	W	E	W	E	W	E
Beijing	0.004	0.117	0.045	0.076	0.132	0.037	0.059	0.026	0.033	0.133
Tianjin	0.002	0.097	0.047	0.056	0.099	0.031	0.051	0.039	0.028	0.129
Hebei	0.012	0.080	0.118	0.063	0.071	0.029	0.063	0.038	0.101	0.074
Shanxi	0.012	0.068	0.042	0.038	0.070	0.020	0.048	0.027	0.110	0.070
Neimenggu	0.069	0.071	0.038	0.038	0.071	0.050	0.070	0.047	0.157	0.081
Liaoning	0.024	0.077	0.059	0.046	0.080	0.033	0.060	0.026	0.100	0.107
Jilin	0.037	0.074	0.055	0.034	0.075	0.039	0.058	0.028	0.081	0.099
Heilongjiang	0.065	0.077	0.129	0.041	0.075	0.057	0.066	0.035	0.116	0.105
Shanghai	0.007	0.118	0.044	0.086	0.116	0.033	0.045	0.023	0.046	0.145
Jiangsu	0.020	0.092	0.146	0.073	0.080	0.032	0.055	0.024	0.036	0.101
Zhejiang	0.053	0.104	0.061	0.058	0.091	0.031	0.056	0.023	0.069	0.091
Anhui	0.034	0.064	0.097	0.056	0.055	0.025	0.053	0.022	0.066	0.060
Fujian	0.082	0.075	0.040	0.038	0.064	0.034	0.058	0.017	0.031	0.080
Jiangxi	0.086	0.061	0.040	0.041	0.054	0.022	0.062	0.019	0.099	0.074
Shandong	0.016	0.081	0.143	0.077	0.074	0.033	0.060	0.034	0.069	0.078
Henan	0.019	0.069	0.113	0.084	0.062	0.028	0.053	0.024	0.076	0.062
Hubei	0.049	0.065	0.071	0.057	0.063	0.030	0.051	0.019	0.076	0.079
Hunan	0.073	0.060	0.057	0.062	0.055	0.026	0.069	0.019	0.083	0.062
Guangdong	0.074	0.076	0.054	0.064	0.068	0.031	0.063	0.017	0.040	0.086
Guangxi	0.098	0.055	0.040	0.046	0.050	0.025	0.075	0.014	0.061	0.048
Hainan	0.078	0.052	0.019	0.020	0.054	0.038	0.075	0.017	0.028	0.075
Chongqing	0.038	0.054	0.030	0.030	0.050	0.024	0.031	0.017	0.082	0.062
Sichuan	0.093	0.055	0.040	0.077	0.051	0.023	0.051	0.012	0.113	0.056
Guizhou	0.056	0.044	0.024	0.037	0.037	0.018	0.049	0.013	0.101	0.034
Yunnan	0.114	0.049	0.040	0.043	0.043	0.022	0.064	0.019	0.104	0.031
Xizang	0.176	0.039	0.002	0.013	0.041	0.061	0.071	0.059	0.077	0.020
Shanxi	0.029	0.069	0.025	0.041	0.067	0.026	0.056	0.019	0.121	0.051
Gansu	0.024	0.055	0.022	0.030	0.052	0.024	0.064	0.022	0.140	0.042
Qinghai	0.079	0.057	0.011	0.022	0.058	0.020	0.066	0.041	0.086	0.053
Ningxia	0.003	0.066	0.014	0.019	0.064	0.032	0.070	0.036	0.072	0.067
Xinjiang	0.107	0.065	0.058	0.044	0.071	0.042	0.080	0.033	0.055	0.061

**Table 6 ijerph-18-01540-t006:** Changing trends of harmonious development (HD) level and lag type in rural China from 1997 to 2019.

SD	1997–2003 Mean	Lag	2004–2011 Mean	Lag	2012–2019 Mean	Lag	1997–2019 Mean	Lag
Beijing	0.479	V	0.501	II	0.510	II	0.496	III
Tianjin	0.419	III	0.480	I	0.516	III	0.469	I
Hebei	0.360	III	0.437	II	0.593	III	0.456	III
Shanxi	0.292	I	0.376	I	0.498	I	0.382	I
Neimenggu	0.281	V	I	I	0.577	III	0.403	III
Liaoning	0.406	III	0.474	II	0.554	III	0.474	II
Jilin	0.314	V	0.368	I	0.491	III	0.385	III
Heilongjiang	0.333	I	0.376	I	0.472	II	0.389	I
Shanghai	0.517	V	0.530	I	0.560	II	0.534	II
Jiangsu	0.445	V	0.603	II	0.634	III	0.556	III
Zhejiang	0.394	V	0.544	III	0.676	III	0.530	IV
Anhui	0.345	IV	0.345	III	0.469	III	0.381	IV
Fujian	0.345	I	0.412	II	0.563	II	0.432	II
Jiangxi	0.273	V	0.327	II	0.438	I	0.340	III
Shandong	0.433	V	0.574	I	0.676	III	0.554	III
Henan	0.355	V	0.426	III	0.579	III	0.446	IV
Hubei	0.329	I	0.390	V	0.548	III	0.415	V
Hunan	0.306	I	0.364	IV	0.501	III	0.384	III
Guangdong	0.483	V	0.656	III	0.743	III	0.621	IV
Guangxi	0.277	I	0.321	V	0.432	III	0.338	V
Hainan	0.255	III	0.308	III	0.391	III	0.314	III
Chongqing	0.277	IV	0.353	II	0.502	III	0.370	III
Sichuan	0.320	V	0.386	I	0.540	II	0.408	III
Guizhou	0.226	IV	0.277	II	0.376	III	0.288	III
Yunnan	0.264	I	0.303	I	0.405	III	0.319	I
Xizang	0.185	IV	0.212	II	0.290	III	0.226	III
Shanxi	0.273	IV	0.352	IV	0.511	III	0.371	IV
Gansu	0.241	III	0.294	III	0.390	II	0.304	III
Qinghai	0.206	0.180	0.272	I	0.386	III	0.282	III
Ningxia	0.215	II	0.293	II	0.418	III	0.302	II
Xinjiang	0.265	0.306	0.308	III	0.407	II	0.322	III

## Data Availability

Data available on request due to restrictions e.g., privacy or ethical. The data presented in this study are available on request from the corresponding author. The data are not publicly available due to the strict management of various data and technical resources within the research teams.

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
