# Peer review of "Spatial-Temporal Relationship between Water Resources and Economic Development in Rural China from a Poverty Perspective"

_ijerph, 2021, doi:10.3390/ijerph18041540_

Round 1

Reviewer 1 Report

Author(s) mostly responded the previous comments. However, methodology and result presentations still need improvement. The impact of the presented results (in tables 4-5-6) for each province, and how the author(s) reach the proposed conclusion are still not clear in the text .

- I suggest author(s) to pick two exemplary provinces. First, explain how the model is applied on these provinces. Then, explain what the results mean on Tables 4, 5 ad 6 for these selected provinces.  

- Grammar mistakes are not corrected in the text. 

Author Response

1- Author(s) mostly responded the previous comments. However, methodology and result presentations still need improvement. The impact of the presented results (in tables 4-5-6) for each province, and how the author(s) reach the proposed conclusion are still not clear in the text. I suggest author(s) to pick two exemplary provinces. First, explain how the model is applied on these provinces. Then, explain what the results mean on Tables 4, 5 ad 6 for these selected provinces. Response: Thank you for this helpful suggestion. About Table 4, in the tracked changes version, Line 301-311, it is that “For example, Guangdong lies in the southeast coastal area. Its water poverty values increased from 0.415 to 0.518, with an average annual growth rate is 1.13%. However, the economic poverty values in Guangdong increased from 0.246 to 0.638, with an average annual growth rate is 7.24%. The economic development speed is much faster than the improvement speed of water resources system. In the inland region, the water poverty in Shaanxi increased from 0.229 to 0.272, with an average annual growth rate is 0.85%, which was almost at a standstill. Policy intervention on water resources system was urgently needed. However, the economic poverty in Shaanxi increased from 0.083 to 0.331, with an average annual growth rate is 13.58%. The economic development speed is much faster than the improvement speed of water resources system, which shows a situation of disharmony between water resources and economic development.” About Table 5, Line 347-354, I have already pointed it out. It is that “For example, in Ningxia, resources and access components of water poverty, human capital, natural capital and physical capital components of economic poverty should be gave priority to address because of their values are the lowest compared to other WPI and SLA components. Also, within resources and use components, it should focus on improve the situation of numbers of reservoir and percentage of rural population with access to clean water, because some of the components, indicators and Variables cannot be managed (e.g. resource variability and availability) [40]. Economic poverty and so on.” About Table 6, Line 369-376, I have revised it. It is that “For example, the H-D level of Beijing is 0.479, and it is strong water lag in 1997-2003; and in 2012-2019, the H-D level of Beijing is 0.496, and it is strong water lag. This shows that the contradiction between economic development and water shortage has been alleviated, however, water shortages remain a hindrance. By contrast, the H-D level of Xizang is 0.185, and it is strong water lag in 1997-2003; and in 2012-2019, the H-D level of Xizang is 0.226, and it is strong economic lag. This shows that the contradiction between economic development and water shortage has been serious. It need to find the reasons for the inharmonious, this allows for policy intervention.”. 2 - Grammar mistakes are not corrected in the text Response: Thank you for this helpful suggestion. I have used English Editing Services of MPDI to revise it. Its number is english-26270.

Reviewer 2 Report

Thank you for working on the comments made earlier and for providing further clarifications. I still find a number of aspects the authors of the paper should improve:

1) the main problem I have with the paper is its relatively unclear (or: dual) focus. From the authors' clarifications I understand now that the main focus of this paper is to develop an analytical approach rather than to present results of an empirical analysis to facilitate policy actions. Yet, the current manuscript actually pretends to deliver both, while it does not fully meet expectations on any of these two purposes. For a methodological paper the current manuscript does not clearly enough show the need (i.e. shortcoming of the current approach), and the advantages brought by the new approach. In L476 the deficiencies of the current index system are referred to, but these have not clearly been stated in the paper. For an empirical paper, discussion and policy implications are too weak, some of the methodological elaborations (e.g., section 3.2) are too extensive, and some of the conclusions are not backed up by the results (e.g., L412 - from which data can this be concluded?). The focus of the entire paper should therefore be revised such that its methodological contribution stands out more clearly, including the discussion and conclusions sections.

2) While the authors have tried to improve the paper based on earlier comments, some questions still remain and the presentation of the paper should be improved in some parts. For example, the introduction should clearly state the problem and objective of the paper. Currently, important aspects related to this are only mentioned in L489-492 - these should be moved into the introduction. The advantage over potential alternatives of using a combination of SLA and WP concepts should be more clearly stated. Sentence starting in L342: It is not easy to see this in Table 5, better clearly state the values this sentence refers to. L362: which "final results"? L373: How have the four categories been divided using SPSS?

3) The results in Table 4, 5 and 6 should be presented in a different, more easily comprehensible way. One way could be "heat maps" that use different color codings for high/low values or positive/negative trends. The "Lag" in Table 6, rather than showing a decimal figure, could be indicated by using the categories I-V as provided in Figure 3.

4) Limitations of the developed approach should be critically discussed. For example, how would results change if other available indicators were used to assess WP or EP?

5) As stated in my previous comments, I would suggest to ommit political statements that do not contribute to the content of the paper. This also concerns the maps (Figures 1, 4 and 5) which consistently show the Islands in the South Chinese Sea as territory of PRC, despite the fact that this claim is internationally disputed. As the analysis does apparently not include these areas anyway, I suggest to remove this part of the maps.

6) Extensive language editing is required throught the entire text. Several sentences show word omissions, wrong transitions, gramatical and/or other language problems (e.g., sentences starting in L55, 157, 160, 251, 309, 333, 335, 342, 344, 345, 348, 350). Many paragraphs do not have a clear topic sentence, hence the reader cannot follow the flow of thoughts easily. The manuscript should be signficantly shortened to make it more appealing to the reader.

Round 2

Reviewer 2 Report

Thank you for further improving the paper. I find that most of my earlier comments have been properly addressed.

In regard to my previous comment 9, I would uphold my suggestion to use different color codes (e.g., dark red...dark green) in Table 5 and perhaps Table 6 to visually distinguish low values from high values.

In addition, I have noted the following minor mistakes: L100: correct into "frameworks that handle"